# The azimuth observation by Global Navigation Satellite Systems as an alternative to astronomical method: A case study at Kakioka

Hiroki Matsushita [1], Daisuke Matsuura[1], Funa Iizuka[1], Junpei Oogi[1], Seiki Asari[1]

[1]Kakioka Magnetic Observatory, Japan Meteorological Agency, Kakioka 595, Ishioka, 315-0116, Japan

*Correspondence to*: Hiroki Matsushita (matsushita_hiro@met.kishou.go.jp)

**Abstract.** For the azimuth observation to be made at its magnetic observatories routinely, Japan Meteorological Agency (JMA) has adopted a traditional method based on Polaris sighting. Due to its difficulty to implement under overcast weather conditions and to its demand on observers, for overtime work into the evening, we are motivated to seek for an alternative method based on the GNSS observation that might potentially relieve those two disadvantages. An experiment is made at Kakioka to assess

the eligibility and effectiveness of the GNSS method for JMA's unmanned observatories, Memambetsu and Kanoya. The GNSS observations themselves achieve as high a precision as approximately 1 arcsecond, as far as they are analyzed with Static mode. Derived from the results of GNSS observation and some supplementary horizontal angle measurements, the azimuth of the azimuth mark for the absolute measurement is determined with a precision of a few arcsecond, which is comparable to the azimuth precision achieved by the Polaris sighting. However, we end up with their significant difference by

about 10 arcseconds. We discuss this discrepancy to be possibly due to a local geoid gradient. The Polaris observation is made with a theodolite tilted in the gravitational direction, also known as the vertical line deviation, whereas the GNSS observations are based on the azimuth of the compliant ellipsoid plane.

## 1 Introduction

The geomagnetic vector field consists of components such as "total magnetic force," "declination angle," "inclination angle,"

"horizontal partial force," and "vertical partial force." The declination represents the deflection of the magnetic north (north pointed to by a magnetic compass) from the true north (the geographic north on a map). Although the true north is more relevant in everyday life, it is usually difficult to sense straightforwardly. As the magnetic north can easily be determined using a compass, the magnetic north is rather useful for indirectly pointing the true north by referring to a given value of the location's declination.

In the case of geomagnetic observations, the azimuth — defined hereafter as the horizontal angle (clockwise positive) of a direction of interest with respect to the true north — must be determined by an independent method. One is the traditional astronomical method using a celestial body at night. In this method, which has been revitalized because of its high accuracy (see Barazzetti, 2025), the horizontal angle between an azimuth mark and a reference star, such as Polaris, is measured. Horizontal and elevation angles of the reference star in the equatorial coordinate system are provided by a celestial orbit

calculation software. The predicted angle can then be combined with the observed angles to calculate the azimuth of the azimuth mark. The same method has also been applied to the sun in many observatories (Jankowski and Sucksdorff, 1996), although it is difficult to achieve a comparable accuracy due to the difficulty of centering of the sun with its much larger appearance and quicker motion.

Alternatively, Global Navigation Satellite Systems (GNSS) can be used to determine the azimuth. By measuring positions at two points by GNSS precisely, the azimuth of one point with respect to the other can be calculated. The positioning accuracy is generally known to be a few millimeters (Static mode), less than a meter (Differential GPS), and more than a few meters (Single mode). Static mode provides the most accurate positioning accuracy; however, it requires extended observation times and two sets of equipment, which increases costs. Single mode is economical since it requires only one piece of instrument, but it has the disadvantage of lower accuracy because systematic errors such as atmospheric delay are typically not corrected. Differential GPS achieves higher accuracy than single mode by obtaining correction data for systematic errors based on Continuously Operating Reference Station data. However, the accuracy of the correction data degrades depending on the distance from the reference station, and in any case, its accuracy is inferior to static mode. In terms of the azimuth, the reported accuracy is about a few arcseconds (e.g. Lalanne et al., 2013). The INTERMAGNET technical reference manual (Bracke, 2025) recently updated suggests to apply Differential GPS, or otherwise the traditional astronomical observation, when redetermination is necessary.

Japan Meteorological Agency has adopted the method using Polaris at its magnetic observatories, Kakioka, Memambetsu and Kanoya, that are registered as the INTERMAGNET observatories (Love and Chulliat, 2013). It has been aware, however, that there are difficulties with this method. First of all, it is subject to weather conditions, with the Polaris sighting being easily precluded by clouds. Particularly, rain clouds make the observation totally infeasible. Furthermore, for an even more accurate observation of Polaris, the optimal season is restricted to late Autumn, when the star migrates vertically rather than horizontally during the typical observation hours in the early evening. These timing limitations make the opportunities of the observation fewer. While this is actually not a major problem at Kakioka which has permanent staff, it significantly matters for Memambetsu and Kanoya, which are unmanned observatories visited by observers only once every two weeks for maintenance work and the absolute measurement. Table 1 presents the number of azimuth observations made at each observatory during the recent observation period from 2022 to 2024. As indicated in the table, while Kakioka conducts approximately 10 observations annually, Memambetsu and Kanoya conduct only a few observations each year. Under the current methodology, increasing these to a level equivalent to Kakioka is difficult. Another difficulty is the extra labor inevitable for the staff to make the azimuth observation with Polaris, for which they have to work overtime after their normal office hours when the sky is sufficiently dark. This requirement is especially burdensome for the staff visiting Memambetsu and Kanoya. There is only one observer for each, travelling all the way there by car accompanied by a supporting member. Obviously, it would be preferable if they had much more chance for the azimuth observations during the daytime.

The Sun-based azimuth method, which is applied by many observatories, can address daytime operation but remains weather-dependent like the Polaris method and is typically less accurate. While methods for improving accuracy have been developed

(e.g., Rasson et al., 2017), achieving the required accuracy of a few arcseconds remains challenging. Practical constraints also arise because observation pillars are typically located indoors and lack windows that enable constant viewing of the sun; our site can view Polaris but not the Sun, so the Sun-based method may be infeasible for such observatories.

Given the circumstances above, we investigate the azimuth observations by GNSS in view of its potential to alleviate the difficulties of our conventional observation. We compare the accuracy of the azimuth observations by GNSS and the Polaris sighting, in order to assess if our conventional observations can eventually be replaced by the GNSS observations, especially at Memambetsu and Kanoya. To this end, we conducted an experiment at Kakioka to investigate general performance of the GNSS approach with reference to our requirement in the precision for the absolute measurement. This paper is organized as follows. In Section 2, starting with an overview of the GNSS observation method, we give detailed description of individual observations and the data processing method. Also, the precision of the azimuth resulting from the GNSS observations is presented. In Section 3, the difference between the azimuth obtained by GNSS and Polaris sighting is discussed. Section 4 summarizes the entire report.

**Table 1: The number of azimuth observations using Polaris from 2022 to 2024**

| Observatory | 2024 | 2023 | 2022 |
|---|---|---|---|
| Kakioka | 7 | 8 | 8 |
| Memambetsu | 3 | 3 | 3 |
| Kanoya | 1 | 1 | 1 |

## 2 The azimuth derived by the GNSS observations

### 2.1 Overview of the whole observation

The entire observation area inside the premise of Kakioka is shown in Figure 1, in which the points of observation and as well as the azimuth mark are also indicated. The observatory's principal absolute pillar, marked as Point C in the figure, is indoors, a precise GNSS observation is hardly possible right at that location and at the azimuth mark. Therefore, it is necessary to have at least two "eccentric points", which are indicated as Point A and Point B in Figure 1. It should be noted that the GSI tiles cited in the figure caption refer to map data disseminated by the Geospatial Information Authority of Japan. The baseline length in between is approximately 150 m. The whole observation consists of the GNSS observations and supplementary measurements of horizontal angles (as schematically illustrated in Figure 2). First, a GNSS observation is performed simultaneously at Points A and B to obtain the azimuth angle $\theta_{GNSS}$ (Section 2.2). Subsequently, horizontal angle observations are made at Points B and C to obtain $\beta_1$ and $\beta_2$, respectively (Section 2.3). The azimuth angle of the azimuth mark, as represented by $\theta_2$ in Figure 2, can generally be derived as (Section 2.4)

$$\theta_1 = \theta_{GNSS} + \beta_1 \pm 180°, \quad (1)$$

$$\theta_2 = \theta_1 + \beta_2 \pm 180°. \quad (2)$$

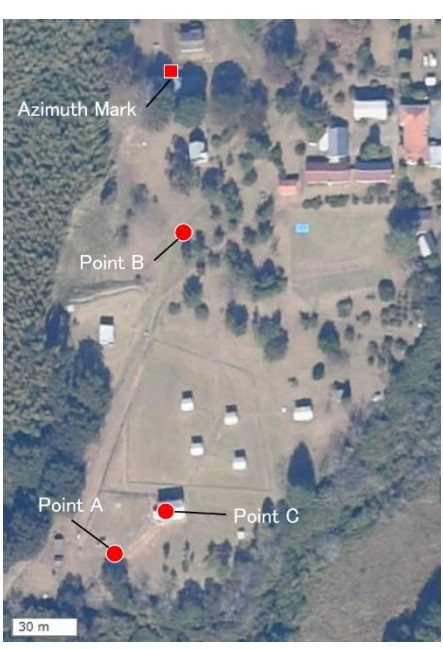

**Figure 1: The area of the observations at Kakioka (Posted with data of points appended to the GSI Tiles).**

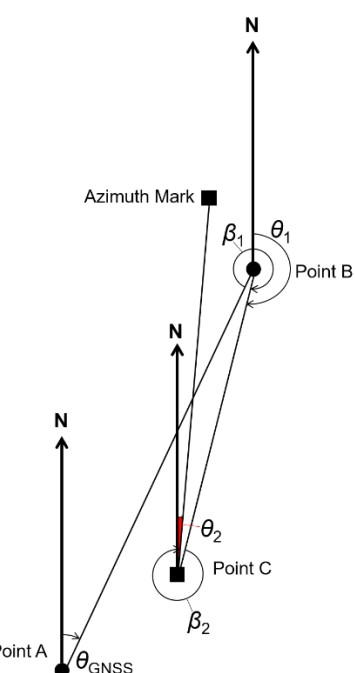

**Figure 2: A schematic diagram of the whole observation.**

## 2.2 GNSS observation and its data processing

GNSS is basically a technique that determines ground-based positions using radio waves from satellites. However, the relative positioning method in GNSS determines the positional relationship between two points (referred to as the baseline vector). The RTKLIB software (Takasu et al., 2007) used for the analysis in this study enables the selection of this baseline vector as an output. Subsequently, the azimuth is calculated from the north-south and east-west components of this vector. The azimuth deviation relative to position is expressed as $\delta\theta = \arctan(\delta y/d)$, where $\delta y$ and $d$ correspond to the deviation of the position

perpendicular to the line of the points and the distance between points, respectively. As indicated in this equation, the effect on azimuth becomes smaller as the distance between points increases.

The observation equipment consists of two GNSS receivers (Trimble Alloy and Trimble R750) and two antennas (both are Zephyr3 Rover) (Figure 3). The measurement is made for an hour at a sampling rate of 1 Hz. The obtained data are then analysed by using RTKLIB ver. 2.4.2 with a setting as summarized in Table 2. In this experiment, the results with Static mode

are employed for the assessment of the azimuth observation by GNSS, because it is widely considered the most accurate (see also our experiment in Appendix A). The L1, L2 and L5 signals are used for the satellite types GPS, GLONASS and QZSS. Because of the short baseline length, neither the atmospheric delay correction nor the ionospheric delay correction is made. The range and standard deviation $\sigma_{\theta GNSS}$ of the azimuth $\theta_{GNSS}$ thus calculated (Figure 4) are found to be less than 10 arcseconds and about 0.4 arcsecond, respectively. As seen in Figure 4, there are several outliers of a few arcseconds. For

instance, at 3:50 and 4:10. These outliers occur primarily when the number of satellites receiving signals changes. Such outliers are excluded during the calculation process.

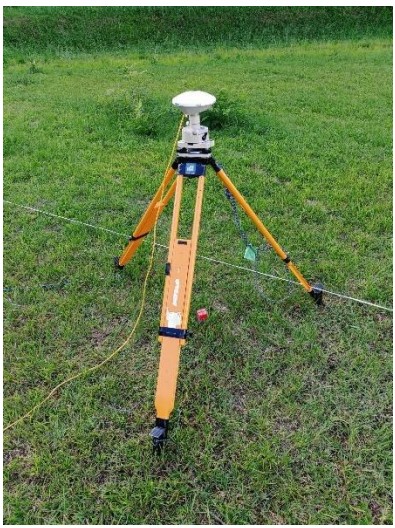

**Figure 3: A photo the GNSS equipment mounted on a tripod.**

**Table 2: The settings of GNSS analysis.**

| Item | Contents |
| --- | --- |
| Software | RTKLIB ver2.4.2 |
| Positioning Mode | Static |
| Types of satellite | GPS, GLONASS, QZSS |
| Frequencies | L1, L2, L5 |
| Satellite Ephemeris | Broadcast |
| Ionosphere Correction | None |
| Troposphere Correction | None |

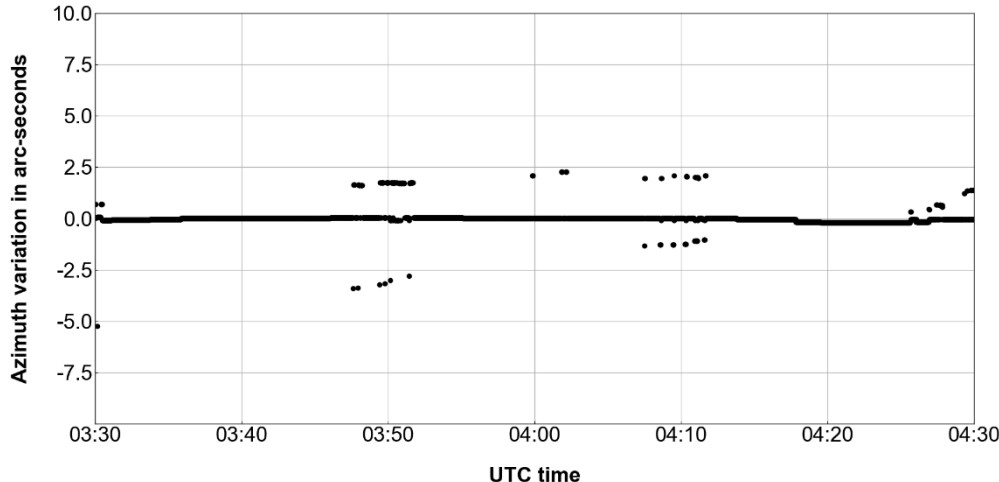

**Figure 4: Observed azimuth $\theta_{GNSS}$ at Point A as a function of time.**

## 2.3 Supplementary measurement of horizontal angles and its processing

For the observations of $\beta_1$ and $\beta_2$, either a theodolite Zeiss (THEO-010B) or a Trimble M3 total station is used (Figure 5). The observations are undertaken basically in compliance with Japanese standard regulations for surveying. To qualify results of those observations, we take 15 and 8 arcseconds respectively as thresholds in terms of the double angle difference and the 130 observation difference. After having acceptable results, extra observations are made repeatedly to verify their accuracy, i.e. 5 times and 6 times for $\beta_1$ and $\beta_2$ (Figure 6), their standard deviations, $\sigma_{\beta 1}$ and $\sigma_{\beta 2}$, being 2.3 and 2.1 arcseconds, respectively.

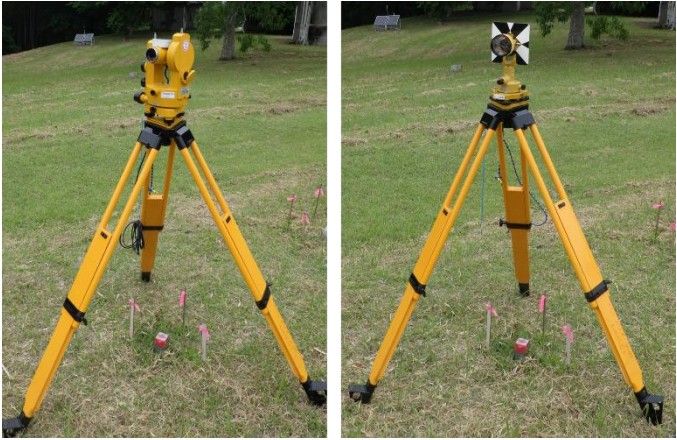

**Figure 5:A set of instruments for the horizontal angle observation. THEO-010B theodolite on the left and a survey prism on the right**

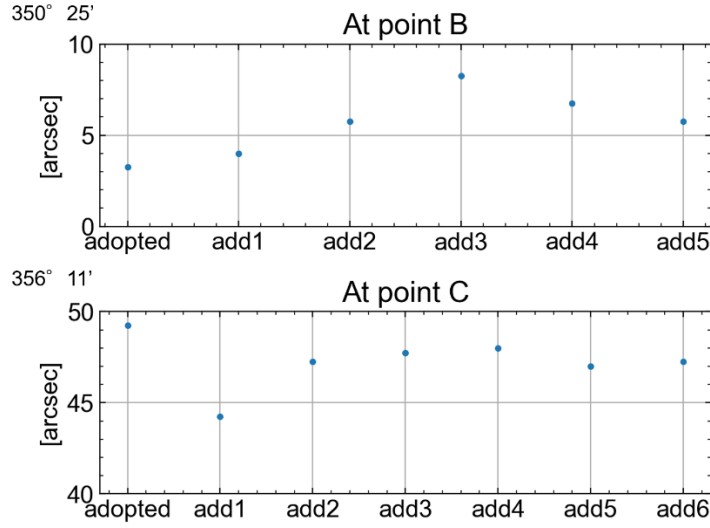

**Figure 6: The adopted results of the horizontal angles, $\beta_1$ at point B and $\beta_2$ at point C, and the results of additional observations.**

## 2.4 Calculation of the azimuth to azimuth mark

The results of individual observations are summarized in Table 3. From these values and equations (1) and (2), the azimuth $\theta_2$ to the azimuth mark at the absolute pillar, or Point C in Figure 1, is calculated to be 0° 32′ 44.6″. Its standard deviation $\sigma_{\theta 2}$ is estimated to be 3.1 arcseconds by applying the propagation law to those of the individual observations while assuming non-correlation among them:

$$\sigma_{\theta 2} = \sqrt{\sigma_{\theta GNSS}^2 + \sigma_{\beta 1}^2 + \sigma_{\beta 2}^2}$$

**Table 3: The observed results of $\theta_{GNSS}, \beta_1, \ \beta_2$ and the azimuth $\theta_2$ calculated from them.**

|  | Value | Standard deviation |
|---|---|---|
| Azimuth $\theta_{GNSS}$ at Point A | 13° 55′ 52.0″ | 0.4″ |
| Horizontal angle $\beta_1$ at Point B | 350° 25′ 3.3″ | 2.3″ |
| Horizontal angle $\beta_2$ at Point C | 356° 11′ 49.3″ | 2.1″ |
| Azimuth $\theta_2$ at Point C | 0° 32′ 44.6″ | 3.1″ |

## 3 Comparison with the azimuth obtained from Polaris observations

To compare the accuracy and precision of the azimuth observations by GNSS and Polaris sighting, let us first introduce our conventional observations at Kakioka. Annually around October, the observations are performed independently by as many observers as ten, where a single observer sights Polaris eight times. Figure 7 shows the azimuth values derived from the Polaris observations at the Kakioka since 1997. The true azimuth obtained from Polaris observation is defined as follows, based on the average of the annual observation results. First, the results for each year are averaged after excluding obvious outliers. Subsequently, a comparison is made between this average value and the previous year's average to ascertain whether a substantial change has occurred. If the value has not changed significantly, we determine that the true azimuth has remained constant from the previous year. The mean value for the year 2024 is estimated to be approximately $0° 32' 53"$. Furthermore, we define the true azimuth from 2011 to 2024 as $0° 32' 55"$. While there are apparently variances in the results among the observers, the standard deviation of the azimuth by Polaris sighting is usually less than three arcseconds.

The current experiment reveals that the azimuth by GNSS ($\theta_2$ in Table 2) is smaller by about 10 arcseconds than that by Polaris sighting. The difference is unlikely attributable to the observation error, considering the high precision of each observation indicated by their standard deviations as small as a few arcseconds.

What is the cause of this significant difference? Here we will consider Deflection of the Vertical (DoV), which is well known in the field of geodetic surveying (see Vittuari et al., 2016; and the references therein). DoV is defined as a deviation angle between the direction of gravity and the geometric normal to the reference ellipsoid's surface at a given location. The measurement of an angle with the traditional instruments, such as total stations and theodolites, is based on local direction of the gravity, while GNSS observations are based on the ellipsoidal normal. As illustrated in Figure 7, azimuths obtained from the Polaris and GNSS observations can differ if the plumb line differs from the ellipsoidal normal.

Atumi (1933) first reported DoV in Japan. According to his investigation at Tsukubasan which is roughly 10 km west of Kakioka (Table 1 of Atumi (1933)), the astronomical latitude and longitude are $36° 13'22.0"$ N and $140° 5'55.0"$ E, respectively, while the geodetic latitude and longitude are $36° 13'22.3"$ N and $140° 5'67.2"$ E, respectively. The east-west component of DoV $\eta$ can be estimated by

$$\eta = \left(\lambda_a - \lambda_g\right) \cos \varphi_g$$

where $\lambda_a$ is the astronomical longitude and $\varphi_g$ and $\lambda_g$ are the geodetic latitude and longitude, respectively. Using the values reported by Atumi (1933), the $\eta$ turns out to be -9.8 arcseconds. The effect of the DoV $\eta$ on the difference $\delta$ between the azimuth by the GNSS and Polaris observations can be estimated by applying the level correction equation (cf. Kakioka Magnetic Observatory, 1987)

$$\delta = b \tan h ,$$

where $b$ is the tilt of the instrument and $h$ is the altitude angle of Polaris (as illustrated in Figure 8b). Assuming that $b$ is represented by $\eta$ (Figure 8a) at Tsukubasan, $\delta$ is about 7 arcseconds, which roughly explains the observed difference between the azimuth by the Polaris and GNSS observations. For a more rigorous verification, observation or calculation of the very

local DoV at Kakioka would be necessary, as the instrument tilt $b$ can still be somewhat different from the DoV at Tsukubasan $\eta$.

Another potential source of systematic error is positioning deviation during the installation of the GNSS receiver. However, this deviation is typically within a few millimeters, which is too small to account for the errors obtained in this study. In addition, although not documented in this paper, a separate observation conducted in the previous year yielded nearly identical systematic errors. Consequently, it is more natural to consider an underlying cause.

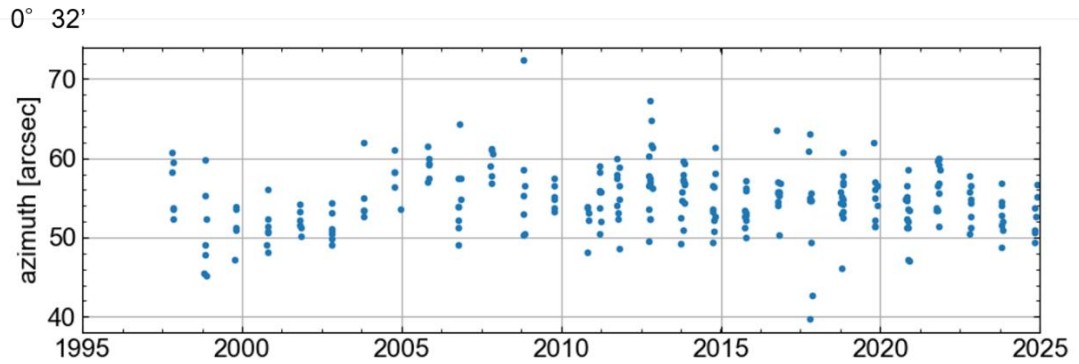

Figure 7: Record of the azimuth by Polaris observation at Kakioka since 1997.

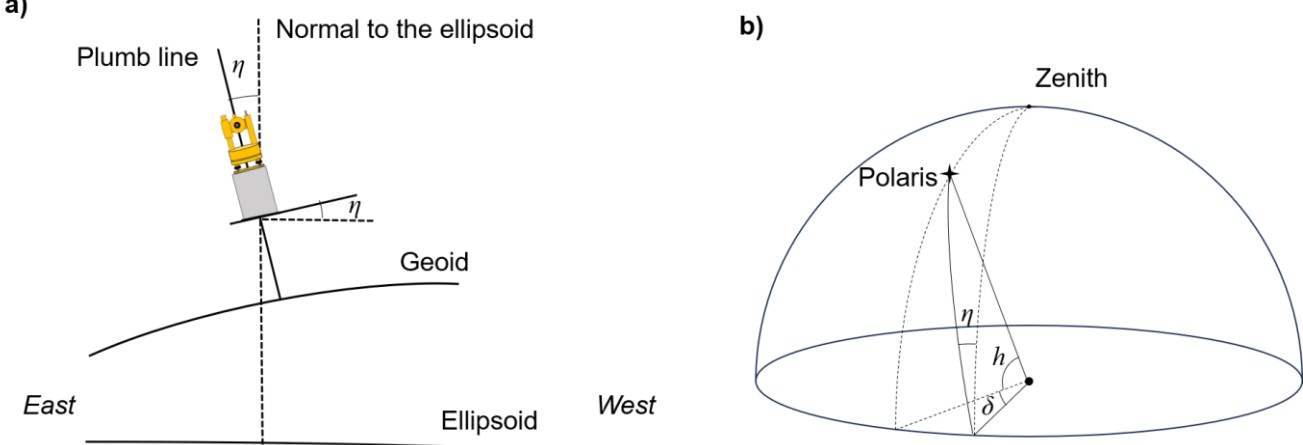

Figure 8: a) An image of DOV in an east-west inclined geoid. b) The deviation $\delta$ in the azimuth, as observed by sighting Polaris at an elevation angle $\eta$ in the presence of DoV by $\eta$.

.

## 4 Summary

While the azimuth observation — the measurement of deflection angle of an azimuth mark with respect to the true north — is essential for the absolute measurement at magnetic observatories, the method with star sighting is associated with difficulties due to weather-dependent observation conditions, as well as demands for overtime work into the evening. A way to mitigate these hardships would be introducing an alternative method with GNSS observation. To examine the applicability of the GNSS technique to the azimuth observations especially at our unmanned observatories, Memambetsu and Kanoya, an experiment is conducted using the GNSS technique at Kakioka in 2024. Under the setting of observation necessarily including horizontal angle measurements at eccentric points (Section 2), the precision of the azimuth of the azimuth mark is found to be a few arcseconds, which is comparable to that derived from previous Polaris observations. A significant difference of about 10 arcseconds is revealed between the two methods. This is due most likely to the DoV, according to the azimuth deviation predicted by using a report of measurement plumb line direction acquired at a distance of roughly 10 km to the west. It indicates a necessity of further investigation, such as an in-situ observation of the DoV. Determining whether the azimuth by GNSS or the one by Polaris is absolutely accurate is difficult. This is because it depends on assumptions about the shape of the Earth. Is it an irregular geoid shape or an ellipsoid? However, when considering geomagnetic modelling, maintaining continuity with past observational results is likely more important. Therefore, when transitioning to the GNSS method, we believe that by determining the DoV at each observatory and adding it to the GNSS results, we can connect to the past Polaris results.

Our transition to GNSS has only recently begun to be considered. In transitioning to the method with GNSS, it will also be necessary to consider even more efficient procedure for these tasks; in this experiment the series of work consisted of an hour of GNSS observations, 30-minute horizontal angle observations, calculations, and preparation of the equipment. In the future, surveys are planned for Kanoya and Memanbetsu, where efficient azimuth observations are of the utmost importance. It is anticipated that the transition to GNSS will commence with these two observatories.

**Appendix A: Comparison of azimuths among different positioning modes**

We introduce some results of the GNSS analysis by choosing different positioning modes. In Figure A1 the positions for different modes are plotted with colored dots on a horizontal plane. Those for Single and Differential GPS (DGPS) modes are relatively largely dispersed, whose ranges are 3 to 5 m and about 1 m, respectively. Figure A2 shows a zoomed-in view for Static and Kinematic modes. Their ranges are about 2 cm and below 1 cm for Kinematic and Static modes, respectively. Figure A3 shows histograms of the azimuth angles $\theta_{GNSS}$ at the main absolute pillar at Kakioka (Point C in Figure 1) converted from equations (1) and (2). Here the horizontal angles, $\beta_1$ and $\beta_2$, are fixed so that the histogram distributions concern only the GNSS observations. The solid black curves illustrate normal distributions representing the mean and standard deviations of the azimuth derived from the Polaris observations in 2024. Obviously, the azimuth derived with Static mode is determined much more precisely than that with the Polaris observation (see also Table 2). The precision provided by Kinematic mode is comparable to the Polaris observation, whereas DGPS and Single modes are much inferior in the precision, and are not suitable as an alternative to the Polaris observation.

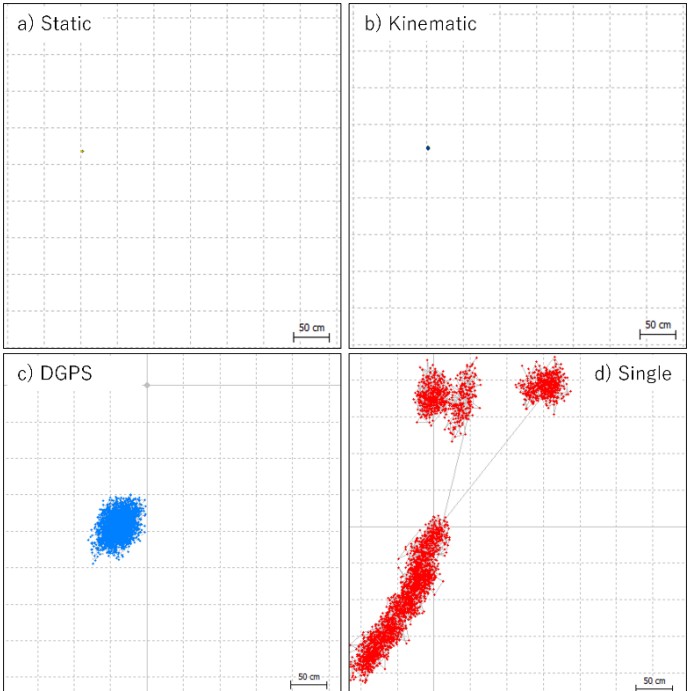

**Figure A1: Distribution of the positions by different positioning modes.**

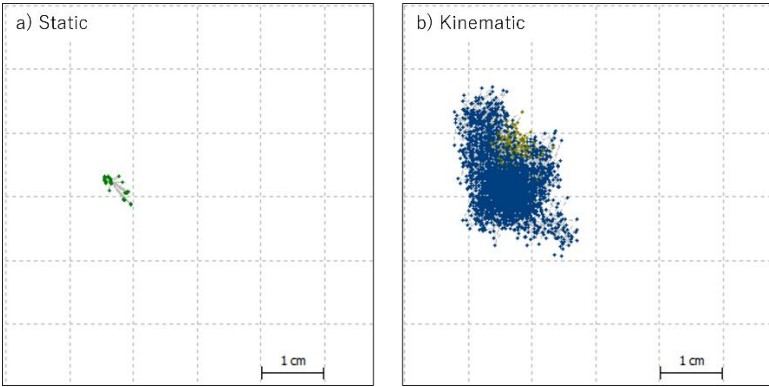

**Figure A2: Zoom-in distribution on the positions for Static and Kinematic modes in the Figure A1.**

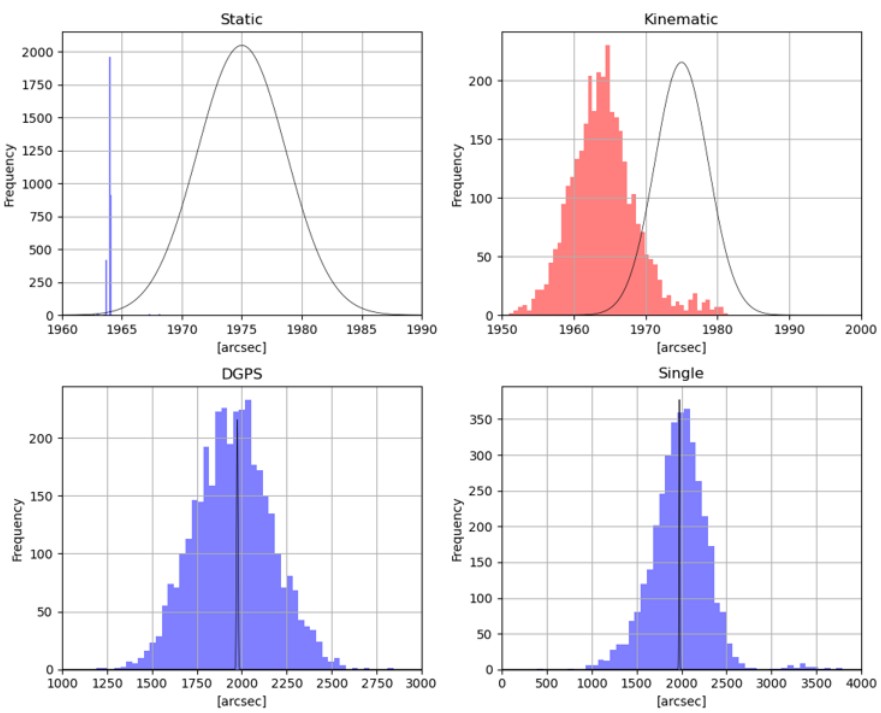

**Figure A3: The histograms of the azimuth by GNSS and the normal distribution of the azimuth by Polaris observation (black line).**

240

Acknowledgements

The authors would like to thank our colleagues for their useful discussions.

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
