# Peer review of "The azimuth observation by Global Navigation Satellite Systems as an alternative to astronomical method: A case study at Kakioka"

_EGUsphere, 2025_

## Author Comment (AC1)

RC1: 'Comment on egusphere-2025-2563', Anonymous Referee #1, 03 Jul 2025

We would like to thank the reviewer for the helpful comments.

The paper by Matsushita et al, "The azimuth observation by GNSS: A case study at Kakioka" is an interesting read. The text is generally easy to follow, and the language is clear. I do not have a background in magnetic observatories, so I can not adequately comment on the novelty of the research. My recommendation is publication following a minor revision.

1. In the title, I suggest that the authors spell out GNSS, so it is immediately clear to all readers, including those outside the magnetic observatory community, what this paper is about by reading the title.

Response:

We agree and are considering the title, "The azimuth observation by Global Navigation Satellite Systems as an alternative to astronomical method: A case study at Kakioka"

2. The introduction is a bit short, and the manuscript contains a very limited number of references. As such, this is not an issue, but I suggest that the author consider expanding the introduction with one or two more paragraphs with background, e.g., the issues that led to this research must likely also be issues at other laboratories.

Response:

We agree with your suggestion and would like to add paragraphs introducing the sunshot method, which uses the sun instead of Polaris. While the sunshot method is also commonly used, it is difficult to apply in the environment of this study.

3. It's a minor issue, but the schematic diagram in figure 1 is not really matched to the actual layout in figure 1. Could the schematic diagram be "twisted" a bit to make the resemblance more intuitive?

Response:

We understood your comment to be pointing out "the schematic diagram in figure 2", not figure 1. We will modify the schematic diagram as follows at the next revision.

4. Figure 4 contains some clear outliers where the signal systematically jumps several arcseconds back and forth. It would be appropriate to include a discussion of the origin of these outliers and how they are handled, e.g., personally, I would be tempted to cull outliers.

Response:

We will include the discussion of these outliers in Section 2.2. Outliers are basically caused by changes in the number of satellites. These outliers are excluded from the calculations. Since they are few in number and not very large, however, they have little effect on the results.

5. The authors find a systematic difference between Polaris observations and GNSS observations. It is suggested that Deflection of the Vertical (DoV) is the main reason for this, and convincing arguments are made. However, it would be appropriate to include a short discussion of any other potential sources for a systematic difference.

Response:

Another potential source is misalignment of the GNSS installation. I would like to include a brief discussion of this topic in Section 3.

6. In the summary of the manuscript, it is not really clear, but still implicitly suggested that JMA is moving towards GNSS based azimuth observations. It would be interesting for the readers to learn a bit more here. Has a decision been made and if so, how will the systematic difference be handled?

Response:

Actually, the decision has not been made since we think the GNSS observation should be performed at also Kanoya and Memambetsu, where the different DoV is assumed. Now that we have gained insight to the azimuth observation by GNSS through this study, we believe we are at the next stage of applying it in Kanoya and Memambetsu.

However, we are glad to provide readers with information about our current situation if they find it helpful. Therefore, we will add an explanation of our future plans to the summary.

---

## Author Comment (AC2)

CC1: 'Comment on egusphere-2025-2563', Thomas Martyn, 29 Aug. 2025

We would like to thank the reviewer for the helpful comments.

The rationale for the paper's study is set out clearly in the Introduction – particularly the difficulties in obtaining regular Polaris measurements. Attaching numerical values to this would improve context e.g. number of visits / measurements annually, number not possible do to cloud cover etc. This could be included in the form of a table and would give improved background.

Response:

We appreciate your suggestion, but we do not officially count days when observations are prevented by cloud cover etc. We have included a table showing the actual number of observations made over the past few years instead.

The description of the process used to calculate the GNSS derived baseline is clear, however as how mentioned by a prior commentor, the schematic could be made to more closely represent the observatory layout.

Response:

The schematic in Figure 2 has been modified.

The fundamental aspect of the study is the ability of GNSS data to accurately determine the observatory azimuth. Much of the discussion, however, focuses on the precision of the measurements obtained with little discussion given to the accuracy of the measurements. To this end, further details of the post processing could be added and their effects / importance discussed. For instance:   the effect of using local Continuously Operating Reference Station to constrain positions (Line 86:   Because of the short baseline length, neither the atmospheric delay correction nor the ionospheric delay correction is made)

Response:

We may not understand the meaning of "the effect of using local Continuously Operating Reference Station to constrain Positions", however, we believe that what is important in calculating the azimuth angle between two points is their relative position. Therefore, we do not think it's necessary to discuss their absolute positions. Additionally, the "accuracy" of the measurement has been discussed in comparisons to Polaris observation. On the other hand, post-processing is summarized in Appendix A. Of the analysis methods compared, static mode analysis showed the highest accuracy. Therefore, we decided to use the static mode in this study.

---

## Author Comment (AC3)

RC2: 'Comment on egusphere-2025-2563', Jan Wittke, 02 Sep 2025

We would like to thank the reviewer for the helpful comments.

The paper by Matsushita et al, "The azimuth observation by GNSS: A case study at Kakioka" describes how to perform GNSS measurements for geomagnetic observatory work.

Please see my comments below:

The paper is well written and the methods as well as the procedures are presented in a concise way. After reading the whole paper, I was wondering about the scientific significance of the approach. Even this might be a relative new way for some geomagnetic observatories, GNSS observations is a common practice in surveying and a well established method. To improve the paper I would suggest following:

- The introduction should cover more the pro and cons of different GNSS survey modes with respect geomagnetic observatory work.

Response:

We have included the pros/cons from the perspective of observation time and the number of instruments used – which is equal to the cost.

- The allover critical value is $\beta_{GSNN}$. There should be a serious discussion how this angle is derived out of the GPS measurements. How does the distance between the points influence the accuracy? How does the calculation is affected with errors?

Response:

We have included more detail of the process of derivation of $\theta_{GNSS}$.

GNSS is basically a technique that determines ground-based positions using radio waves from satellites. However, the relative positioning method in GNSS determines the positional relationship between two points (referred to as the baseline vector). The RTKLIB software utilized for the analysis in this study enables the selection of this baseline vector as an output. Subsequently, the azimuth is calculated from the north-south and east-west components of this vector.

The azimuth deviation relative to position is expressed as the arctangent of the ratio of the position deviation perpendicular to the line of the points to its distance. Here, letting these be $\delta\theta$, $\delta y$, and $d$, respectively, the equation can be expressed as follows: $\delta\theta = \arctan(\delta y/d)$. As can be seen from this equation, the effect on azimuth becomes smaller as the distance between points increases.

- As the angles $\beta_1$ and $\beta_2$ are measured with a theodolite are they also affected by DoV? When yes, how does this influence affect the overall angle $\theta_2$ ?

Response:

There is almost no effect of DoV.

The effect of DoV on azimuth is most significant when the target is at the zenith and becomes zero on the same horizontal plane. In this theodolite measurement, targets and theodolite can be considered to be on nearly the same horizontal plane, so the effect is negligible.

- In the comparison between Polaris observations and GNSS observations, there should be a summary on the Polaris observation method and how uncertainties are compared to the GNSS

method. As the authors identify a potential derivation between both methods it remains unclear which one is more absolute accurate. Is it possible to correct the more inaccurate value with the other method?

Response:

In reality, the azimuth values derived from Polaris observations are defined based on observational results. The inaccuracy referred to here is simply the difference between the defined values and the GNSS results.

Roughly speaking, Polaris's azimuth definition is as follows: if the average of the observational results does not deviate significantly from the previous year's average, the previously defined value is retained. If there is a significant deviation, a new value is redefined after the investigation of its cause. In the past, an earthquake caused the azimuth mark's pillar to shift.

With respect to potential derivation, it is difficult to determine which method is incorrect, as this depends entirely on the assumed shape of the Earth. Is it geoid shape   like bumpy or rotating ellipsoidal. However, the modeling of the geomagnetic field typically assumes a sphere or ellipsoid, we would like to say that the GNSS method is more absolute accurate. But, it should be noted that continuity with past data and consistency with other observatories must also be considered, making it quite complicated.

Yes, it is possible to correct one to another. The simplest way is to add the difference obtained from both methods. Determining the DoV by measuring the local geoid is also possible.

We have included these points in the text.

---

## Editor Decision (ED1)

**Egusphere-2025-2563-manuscript-version 2**

I would like to thank the three reviewers for their dedication in giving comments and suggestions to improve the quality of this manuscript.

The authors have improved the quality of the manuscript by attending successfully to comments and suggestions from reviewers. However, there are minor comments and suggestions for the readability of the manuscript to be attended to before the publication.

**Spelling mistakes**

Line 25: Replace "horizonal" with "horizontal".

**Suggestions**

Line 83: Replace "..., a precise GNSS observation is hardly available right at that location. So is hardly possible right at the azimuth mark." with "..., a precise GNSS observation is hardly possible right at that location and at the azimuth mark."

Line 94: Caption of Figure 1 has words in brackets that are not very clear. The area of the observations at Kakioka (**Posted with data of points appended to the GSI Tiles**). The words in brackets may be rephrased and/or expanded to inform the reader with clarity. GSI Tiles are not defined.

Line 104: Replace "the distance of points" with "the distance between points".

Line 2014: Replace "the number of receiving satellite signals changes" with "the number of satellites receiving signals changes".

Line 126: Replace "a total station (Trimble M3)" with "a Trimble M3 total station".

Line 132: Figure 5 shows two instruments. It would be more informative if these two instruments were specified with their names in the caption (e.g., A on the left and B on the right).

Line 136: Replace "$\beta_1$ and $\beta_2$" with "$\beta_1$ at point B and $\beta_2$ at point C".

Line 162: Replace "Measurement" with "The measurement".

Line 195: Check the use of em dashes, a single em dash might be grammatically correct.

Line 202: Replace "arcsecond" with "arcseconds".

Line 203: Replace "A significant difference of about 10 arcseconds is revealed between those resulting from the two methods" with "A significant difference of about 10 arcseconds is revealed between the two methods".

Line 206: Replace "absolute" with "absolutely".

Line 208: Replace "to GNSS" with "to the GNSS method".

Line 217: Replace "mode" with "modes".

---

## Author Response (AR2)

Egusphere-2025-2563-manuscript-version 2

I would like to thank the three reviewers for their dedication in giving comments and suggestions to improve the quality of this manuscript. The authors have improved the quality of the manuscript by attending successfully to comments and suggestions from reviewers. However, there are minor comments and suggestions for the readability of the manuscript to be attended to before the publication.

Again, thank you for your management of the manuscript.

**Spelling mistakes**

Line 25: Replace "horizonal" with "horizontal".

Replaced.

We also noticed another spelling mistakes on the caption of table 2, and has been modified.

**Suggestions**

Line 83: Replace "···, a precise GNSS observation is hardly available right at that location. So is hardly possible right at the azimuth mark." with "···, aa precise GNSS observation is hardly possible right at that location and at the azimuth mark."

Replaced.

Line 94: Caption of Figure 1 has words in brackets that are not very clear. The area of the observations at Kakioka (Posted with data of points appended to the GSI Tiles). The words in brackets may be rephrased and/or expanded to inform the reader with clarity. GSI Tiles are not defined.

An explanation of GSI Tiles has been added at line 85.

Line 104: Replace "the distance of points" with "the distance between points".

Line 114: Replace "the number of receiving satellite signals changes" with "the number of satellites receiving signals changes".

Line 126: Replace "a total station (Trimble M3)" with "a Trimble M3 total station".

Replaced.

Line 132: Figure 5 shows two instruments. It would be more informative if these two instruments were specified with their names in the caption (e.g., A on the left and B on the right).

Modified.

Line 136: Replace "$\beta 1$ and $\beta 2$" with "$\beta 1$ at point B and $\beta 2$ at point C".

Line 162: Replace "Measurement" with "The measurement".

Replaced.

Line 195: Check the use of em dashes, a single em dash might be grammatically correct.
Modified.

Line 202: Replace "arcsecond" with "arcseconds".
Line 203: Replace "A significant difference of about 10 arcseconds is revealed between those resulting from the two methods" with "A significant difference of about 10 arcseconds is revealed between the two methods".
Line 206: Replace "absolute" with "absolutely".
Line 208: Replace "to GNSS" with "to the GNSS method".
Line 217: Replace "mode" with "modes".
Replaced.